# Cooperative Multi-UAV Collision Avoidance Based on a Complex Network

**Yang Huang, Jun Tang * and Songyang Lao ***

College of Systems Engineering, National University of Defence Technology, Changsha 410073, China;
huangyang13@nudt.edu.cn
* Correspondence: jun.tang@e-campus.uab.cat (J.T.); laosongyang@nudt.edu.cn (S.L.)

**Abstract:** The conflict resolution problem in cooperative unmanned aerial vehicle (UAV) clusters sharing a three-dimensional airspace with increasing air traffic density is very important. This paper innovatively solves this problem by employing the complex network (CN) algorithm. The proposed approach allows a UAV to perform only one maneuver—that of the flight level change. The novel UAV conflict resolution is divided into two steps, corresponding to the key node selection (KS) algorithm based on the node contraction method and the sense selection (SS) algorithm based on an objective function. The efficiency of the cooperative multi-UAV collision avoidance (CA) system improved a lot due to the simple two-step collision avoidance logic. The paper compares the difference between random selection and the use of the node contraction method to select key nodes. Experiments showed that using the node contraction method to select key nodes can make the collision avoidance effect of UAVs better. The CA maneuver was validated with quantitative simulation experiments, demonstrating advantages such as minimal cost when considering the robustness of the global traffic situation, as well as significant real-time and high efficiency. The CN algorithm requires a relatively small computing time that renders the approach highly suitable for solving real-life operational situations.

**Keywords:** Multi-UAV; collision avoidance; complex network; key nodes; robustness; connected component

## 1. Introduction

Collision is an inherent problem in unmanned aerial vehicle (UAV) systems [1–6]. Path planning is important in performing different tasks. Therefore, the demand for the collision avoidance (CA) of UAVs has emerged. With increasing UAV cluster densities, it has become more difficult to perform multi-UAV collaborative tasks, especially if collision avoidance requirement cannot be met. Therefore, the effective planning of flight trajectories and collision avoidance among UAV flight trajectories has become critical in UAV clusters for safe and smooth operation [7].

Various conflict detection and resolution (CDR) theories with different techniques have been proposed to solve this problem. In conventional aviation, several widely used CA systems exist. For example, the traffic alert and CA system (TCAS) is recognized worldwide as an airborne collision avoidance system, and it is the last line of the CDR approach. Technological development has enabled the increase in air traffic density to be mitigated [8,9]. Many studies regarding UAV collision avoidance have been performed in recent years [10–12]. CDR and the optimization of different resolutions have been discussed widely by many researchers and practitioners. These methods can be classified into two primary types of algorithms: Geometric and path planning. Geometric algorithms focus on analyzing the relationship between UAVs and intruders within a specific space to implement passive CA operations. Meanwhile, path planning algorithms focus on obtaining a path for an optimal

connection between the start and end points with minimal security constraints [13–15]. Dimensionality, direction selection, the number of participants, and resolutions are four key elements that should be considered for a UAV's CDR. A detailed summary of publications regarding the different methods of CA is provided in Table 1.

Geometric algorithms exploit state information such as the location, velocity, and the heading of UAVs and intruders to geometrically perform the CA operation. Park, Oh, and Tahk (2008) [16] demonstrated a typical example of this approach, in which a single resolution maneuvering logic called 'vector sharing resolution' was proposed. This method calculates the subtraction of two UAVs' movement vectors in a two-dimensional space and adjusts the trajectories of the UAVs based on the shortest distance vector obtained. Another instant of this method can be found in [17] by Chakravarthy et al. In contrast, this paper solves the problem of obstacles being irregular objects. A collision cone approach is employed by which the irregularly shaped moving objects can be modelled through general quadric surfaces and dynamic inversion-based avoidance strategies can be derived. The limitations of this type of method is the need for information from the intruder UAV, as well as sensitivity to noise from the input data of the sensors.

Path planning algorithms include sampling-based, artificial heuristic, numerical optimization, and decoupled path planning approaches. Each method exhibits its own strength over others in certain aspects. The philosophy of sampling-based approaches is transforming the path-planning problem to obtain a suitable problem from a limited quantity of candidates that is a continuous state space. Lin and Saripalli (2017) [18] presented a method based on the closed-loop rapidly exploring random tree algorithm and three variations of it. Artificial heuristic approaches are highly advantageous during the optimization process, and typical methods include the genetic algorithm [19], particle swarm optimization [20], and artificial bee colony optimization [21]. Numerical optimization approaches rely on some geometric trajectory calculations and aim to obtain an optimal path with respect to a specific objective in the solution sense. The relevant methods include mixed integer linear programming [22], nonlinear programming [23], Dynamic programming [24], quadratic programming [25], and Pontryagin's minimum principle [26]. In decoupled path planning approaches, the basic principle involves obtaining an optimal trajectory under the specific constraints in the first phase; subsequently, the trajectory has to be steered out of the dangerous region. The A*(A-Star) search algorithm was applied to obtain a solution in [27]. The Coronoid graph can be used to generate an initial path that can be subsequently refined by curve fitting with curvature and other constrains [28]. The concepts of space–time reachability of aircraft and space–time potential conflict space were proposed in [29] to consider the uncertainty of the pilot intent. The conflict resolution problem was modeled as a constraint optimization problem in [30,31]. Based on the Single European Sky Air Traffic Management Research and Next Generation Air Transportation System initiatives, [32] proposed a new separation management service to shift the centralized air traffic control interventions to more efficient decentralized tactical operations based on a surrounding traffic analysis tool.

The rapid development of real-time communication technology has made real-time communication between UAVs possible. Therefore, this paper was able to consider using the real-time information of UAV to prevent collision. It is a great idea to use a complex network to analyze the nature of a system. Networks are everywhere, including the internet, electric power grids, supply chains, urban road networks, and the world trade web [33]. The work already done has mainly been about preventing networks from being destroyed [33]. The collision avoidance network is exactly the opposite of most networks, and it is necessary to destroy the network as soon as possible. The collision avoidance based on a complex network (CACN) model grounds the generation of optimal direction changes of the objective function, which integrates the number of intruder UAVs of designated direction, the robustness of the network, and the connected components of the network. The CACN strategy can reduce the domino effect to a certain extent thus improve computational consuming time. In particular, the generated points by the CACN model are recorded and can be further analyzed.

**Table 1.** Summary of publications addressing the CDR problem.

| Publication | Dimension | Maneuvers | Multiple | | Resolution | | | | | | |
| | | | | | Geometric Algorithm | | | Path Planning Algorithm | | | |
| | 3D | T/V/S/C | Pairwise | Global | CPA | CCA | Sampling-Based Path Planning Approaches | Artificial Heuristic Approaches | Numerical Optimization Approaches | Decoupled Path Planning Approaches |
| Park, Oh, and Tahk (2008) | X | C(T,S) | X | | X | | | | | |
| Chakravarthy, Animesh, and Ghose (2012) | X | C(T,V,S) | X | | | X | | | | |
| Lin and Saripalli (2017) | X | C(T,V) | X | | | | X | | | |
| Pehlivanoglu (2012) | X | C(T,V) | | X | | | | X | | |
| Karimi and Pourtakdoust (2013) | X | C(T,V) | | X | | | | X | | |
| Xu, Duan and Liu (2010) | | C(T,V) | | X | | | | X | | |
| Campbell, Bragg and Neogi (2013) | X | C(T,V) | X | | | | | | X | |
| Tisdale, Kim and Hedrick (2009) | | V | X | | | | | | X | |
| Jorris and Cobb (2008) | | C(V,S) | X | | | | | | X | |
| Grancharova et al. (2014) | X | C(T,V,S) | | X | | | | | X | |
| Sridhar, Ng and Chen (2011) | | V | | X | | | | | X | |
| Kim, Gu and Postlethwaite (2008) | X | V | | X | | | | | | X |
| Dai and Cocharan (2010) | | V | | X | | | | | | X |
| Hao, Cheng and Zhang (2018) | X | | X | | | | | | | |
| Yang, Yin and Shen. (2018) | X | | | X | | | | | X | |
| Yang et al. (2019) | | | | X | | | | | X | |
| Radanovic et al. (2018) | X | | X | | X | | | | | |

Note: T = Turns; V = Vertical maneuvers; S = Speed; C() = Combined maneuvers.

The layout of this paper is as follows. Section 2 describes the CA system architecture; Section 3 illustrates the key node selection algorithm and the sense selection algorithm; Section 4 shows the simulation experiments and analyzes the experimental results; finally, conclusions and prospects for future research are summarized in Section 5.

## 2. The Proposed Collision Avoidance System Architecture

Many algorithms in the collision avoidance system support its operation. The core step of the system is to process the information of each trajectory input, which is the four-dimensional (three-dimensional (3D) position + time) information of each UAV, and an analysis on whether any threat is present in the next time interval. If a threat is detected, the system immediately reacts to calculate a suitable CA strategy. The instructions are subsequently communicated to each UAV to synergistically resolve the current threat.

The distributed dynamic optimization approach (DDOA) is used to generate suitable direction changes for the UAVs such that the problem of CA can be mitigated. The core of this approach consists of two parts: The conflict detection (CD) module is used to forecast the threats, and the conflict resolution (CR) module is applied to resolve the detected conflicts. The complex network (CN) aims to avoid collisions by choosing the suitable UAV and the corresponding UAV with a local optimization scope. The CN algorithm only allows for one UAV to perform conflict resolution at one time, which is the principle of DDOA.

## 3. Core Algorithm Analysis

This section describes the complex network model for UAV collision avoidance system and the collision avoidance algorithm in detail. The collision avoidance algorithm consists of two parts: The key node selection algorithm and the collision avoidance direction-selection algorithm.

### 3.1. Representation of Cooperative UAV Flight and Conflict Detection

The background of the problem is that when large scale UAV clusters meet, it is very likely that conflicts will occur. For simplicity, weather perturbations are not discussed when analyzing the trajectories. UAVs share a common airspace in the same or different flight levels, and a UAV cluster can be considered safe only if the separation between each UAV is larger than the given safety distance. A sequence of discrete points is used to model the trajectories of the UAV, and speed information can also be obtained from these discrete tracking points.

Because UAV clusters frequently experience threats, and some threats are regarded as potential threats, it is more efficient to perform one resolution at a time rather than comprehensively considering each threat. By constructing a complex network of UAV groups and analyzing the nature of the network under threat conditions to determine the appropriate CA strategy, the human–machine group can self-organize CA under the condition of performance constraints, thus significantly improving the real-time performance and applicability of CA in UAV clusters.

The key node in the complex network theory exhibits important properties [33–36], and it exhibits the greatest effect on the entire complex network system. The UAV CA algorithm based on the complex network proposed herein considers the safety indicators of the UAV group system at each moment, such that the overall safety of the entire flight process is maximized. The CA system consists of two key algorithms: A key node selection algorithm and a collision direction-selection algorithm. The algorithm expresses the threat between the UAVs using a complex network. The UAV network model is established based on three parameters: Flight speed, flight angle, and security zone. In a real-time communication environment, all UAVs fly toward a given target and must perform CA operations. Each UAV owns a flight trajectory control module, and the CA resolution calculated by the CA system interacts with the surrounding UAVs through real-time data links.

The UAV node is used to indicate that the state of the UAV is monitored by the system in real time. Its attributes are expressed as {UAV, velocity, position, angle, t, state, sense, strength, approaching

time, time cost, key-node}. The UAV indicates the UAV number, velocity indicates the UAV speed, position indicates the current position of the UAV, angle indicates UAV flight angle, t indicates the current time of the UAV, state indicates the UAV flight status, 'sense' indicates that the UAV collision direction is selected, 'strength' indicates the change direction of the UAV, 'approaching time' indicates the remaining time of the UAV approaching the nearest point, 'time cost' indicates the entire flight time consumption, and 'key-node' indicates whether the UAV was selected as a key node.

To detect collisions, UAVs are identified in the Cartesian system. The motion of each UAV is driven by the following:

$$\vec{R}_i(t) = \begin{bmatrix} \vec{r}_{x,i}(t) \\ \vec{r}_{y,i}(t) \\ \vec{r}_{z,i}(t) \end{bmatrix}, V_i(t) = \dot{\vec{R}}_i(t) = \begin{bmatrix} \dot{\vec{r}}_{x,i}(t) \\ \dot{\vec{r}}_{y,i}(t) \\ \dot{\vec{r}}_{z,i}(t) \end{bmatrix} = \begin{bmatrix} \vec{v}_i(t) \cos \varphi_i^t \cos \theta_i^t \\ \vec{v}_i(t) \cos \varphi_i^t \sin \theta_i^t \\ \vec{v}_i(t) \sin \varphi_i^t \end{bmatrix} \tag{1}$$

In the above formula, $\vec{R}_i(t)$ and $V_i(t)$ are defined as the position vector and speed vector, respectively, of $UAV_i$ at time $t$. The foot mark $i$ represent $i$-th UAV. The subscript indicates the horizontal axis system and refers to the height. Let $\theta_i^t$ designate the horizontal angle which is the direction of speed $V_i(t)$ in the $x - y$ plane (counter-clockwise along the $x$ axis). Let $\varphi_i^t$ represent the angle of climb, which is the direction of speed $V_i(t)$ in the vertical plane (measured from the $x - y$ plane, with positive being the upwards direction and negative being the downwards direction).

The maximum climb (tilt) angle, which is determined by the performance of the UAV, is defined to limit the maximum angle at which the UAV will rise and fall in the vertical plane when performing a CA mission, thus setting the maximum pitch change angle. The constraint can be expressed as follows:

$$\varphi_{i,\min} \le \varphi_i(t) \le \varphi_{i,\max} \tag{2}$$

The collection of points in the next time step is as follows:

$$P(i) = \{s \in V | \|\vec{R}_i - \vec{R}_s\| = v \cdot \Delta t; \varphi_{\max} = \arg \sin(|z_i - z_s| / \|\vec{R}_i - \vec{R}_s\|)\} \tag{3}$$

Note that $z_i$ and $z_s$ are defined as the current attitude of $UAV_i$ and the candidate attitude of $UAV_i$ in the next time step, respectively. The symbols $v$ and $\Delta t$ refer to the absolute velocity of UAVs and time interval of each time step, respectively.

When the first pair of UAVs involved in the conflict is detected, each UAV maintains its speed before the first pair of UAVs reaches the danger point to simplify the conflict situation. To determine if a threat of conflict exists, the scope and vertical criteria must be met. This time interval is defined as the time closest to the point of approach (CPA) on the horizontal plane:

$$T_{h,ij}^t = \frac{\left| \vec{r}_{h,i}^t - \vec{r}_{h,j}^t \right|}{\left| \vec{v}_{h,i}^t - \vec{v}_{h,j}^t \right| \cdot \cos(\alpha_{ij}^t - \beta_{ij}^t)}, (\vec{r}_{h,ij}^t \cdot \vec{v}_{h,ij}^t < 0) \tag{4}$$

$$\alpha_{ij}^t = \arctan(\left| \vec{r}_{x,ij}^t \right| / \left| \vec{r}_{y,ij}^t \right|) \tag{5}$$

$$\beta_{ij}^t = \arctan(\left| \vec{v}_{x,ij}^t \right| / \left| \vec{v}_{y,ij}^t \right|) \tag{6}$$

The formula defines $T_{h,ij}^t$ as the time to CPA in the case of a horizontal plane between $UAV_i$ and $UAV_j$ at time $t$. The subscript $h$ indicates the horizontal plane, while $i$ and $j$ indicate the serial numbers

of the UAVs. The equation is defined on the condition that the denominator is not equal to zero. The following equation is defined as the time CPA in the case of a vertical plane:

$$T_{z,ij}^t = \frac{\left|\overrightarrow{r}_{z,ij}^t\right|}{\left|\overrightarrow{v}_{z,ij}^t\right|}, (\overrightarrow{r}_{z,ij}^t \cdot \overrightarrow{v}_{z,ij}^t < 0) \tag{7}$$

A traffic alert (TA) event will be triggered if the following conditions are satisfied:

$$(0 < T_{h,ij}^t < T_{TA}) \wedge (0 < T_{z,ij}^t < T_{TA}) \wedge (D_{h,ij}^{t+T_{h,ij}^t} < DMOR_{RA}) \wedge (D_{z,ij}^{t+T_{h,ij}^t} < ZTHR_{RA}) \tag{8}$$

$DMOR_{RA}$ and $ZTHR_{RA}$ are the range and altitude limit values for safe aviation, respectively, in case of the appearance of slow-closure-rate encounters which make the time threshold values not feasible. $D_{h,ij}^{t+T_{h,ij}^t}$ and $D_{z,ij}^{t+T_{h,ij}^t}$ are the horizontal and vertical distances between $UAV_i$ and $UAV_j$ at CPA, respectively. We used the time when the first pair of $UAV_s$ triggers a TA alert. From that moment, we built a network where each UAV is represented as a node. If a pair of nodes collides, the node will establish a connection. At that moment, all nodes in the system will verify if they are approaching; if they are approaching each other, the two nodes are connected:

$$(\overrightarrow{r}_{h,ij}^t \cdot \overrightarrow{v}_{h,ij}^t < 0) \wedge (\overrightarrow{r}_{z,ij}^t \cdot \overrightarrow{v}_{z,ij}^t < 0) \tag{9}$$

In the case of a pair of UAVs, CA is based on the following principles. As shown in Figure 1, $UAV_1$ cruises from the right side to the left side, whereas $UAV_2$ and $UAV_3$ cruise from the left side to the right side. At this time, $UAV_1$ is selected as the key node. According to the scene at this time, the direction of the CA direction and the target point at the time of the predicted CPA choosing an upward climb or a downward climb may be selected. Furthermore, $UAV_2$ and the $UAV_3$ must maintain the minimum safety altitude-limitation. The specific key node selection methods and direction selection strategies are discussed later.

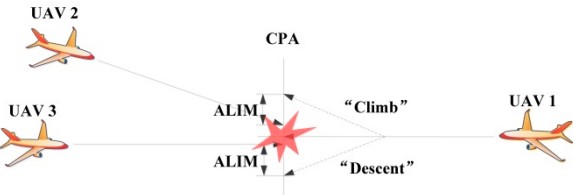

**Figure 1.** Traffic alert and CA (collision avoidance) system (TCAS) coordination for multi-unmanned aerial vehicles (UAVs).

### 3.2. Key Node Selection

The first problem to solve in an encounter is choosing a key UAV to amend its trajectory to avoid collision. The method proposed herein decides the priority of the UAVs via the analysis of the network formed in the collision space. The most threatened one is chosen as the key node. Obviously, the UAV with the higher priority performs collision avoidance operations first.

To clarify, it is necessary to define the collision space. When the system detects a pair of UAVs exhibiting a collision risk, the system obtains all the possible UAV numbers to be hit in the future through global data analysis, and then it obtains the status of the corresponding UAV. The airspace formed by these UAVs is called the collision space. Among them, a connection exists between UAVs that are at the risk of collision, and a network is formed for analysis. An adjacency matrix is applied to record the status between the UAVs. The node represents the UAV, and the edge represents the

relationship between the UAVs. A set is defined to contain the nodes that satisfy the condition. As shown in Figure 2, the nodes represent the corresponding UAVs.

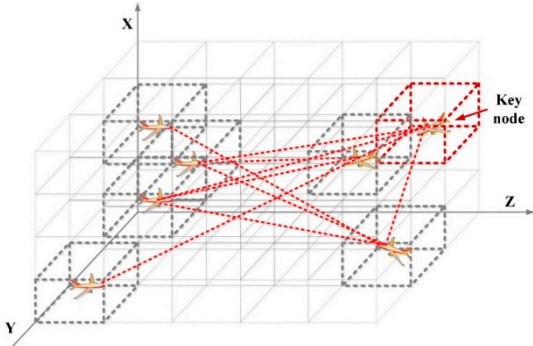

**Figure 2.** Conceptual graph for key node selection.

After the network is completed, a network description is established to describe the internal characteristics of the multi-UAV system. The nodes in the network exhibit different properties, because the position and velocity are different and the degree of threat is different. Thus, the key node can be chosen according to its properties.

First, the easiest method to measure the degree of threat of the UAVs is by examining the situation of the connected edges in the network; this is called the key node selecting method based on degree. To render the network closer to the real scene, it is insufficient to only use the number of edges. The degree approach requires more detailed information, and the edge is a good choice. The edges should contain extra properties to monitor the true relationship of the UAVs. The relationship includes the distance and relative speed between UAVs. The closer a pair of UAVs is to each other, the more dangerous they are. Therefore, the weight of each side depends on the relative distance and relative speed between the UAVs, which is defined as the edge weight between the UAVs:

$$\omega_{ij} = \frac{\vec{v}_{ij} \cdot \vec{d}_{ij}}{\left|\vec{d}_{ij}\right| \cdot \left|\vec{d}_{ij}\right|} \tag{10}$$

where $\vec{v}_{ij}$ is the relative speed of the vector direction, and $\vec{d}_{ij}$ represent the distance between the UAVs. This implies that the faster the two UAVs approach each other, the greater is the weight of the edge between the two UAVs. The degree of importance of each node can be based on the weight of all edges. The summed value represents the degree of importance of the node.

There are many other methods to rank the importance of nodes [34,37]. Another improved method based on node contraction [38] proposed herein aims to excavate the deeper threat of the UAV networks. There are three reasons for using the node contraction method in selecting the key node: First, the principle of key node selection is to fully exploit the importance of the node, for which the network cohesion is a good measure. The choice of key nodes is not only related to the current degree, it is also to the location of the node. If the node is in the fortress position, shrinking the node will result in a much smaller average path length of the network, resulting in greater network cohesion. This method to obtain key nodes is consistent with the key node selection requirements of the UAV group in performing CA operations, i.e., to determine the current and potential threats of the drone to perform CA. Second, the node contraction method has a shorter calculation time than the node deletion method and the median method, and it does not exponentially increase when the number of nodes is small. Finally, the original method of selecting the key nodes has its limitations, and the changes in the structure of the cluster cannot be detected well. Therefore, this method is not considered for some structurally important points.

**Notation**

**Sets:**

$U = \{u_1, u_2, u_3, \ldots, u_n\}$ sets of node representing UAV flying in the air sector under control.

$C = \{c_1, c_2, c_3, \ldots, c_n\} \subseteq U \times U$, sets of edge implying pairs of UAV that are potentially in conflict.

$V = \{\vec{v}_1, \vec{v}_2, \vec{v}_3, \ldots, \vec{v}_n\} \subseteq U \times U$, sets of relative velocity between UAVs.

$G = (U, C)$, graph representing the complex network of the UAV group.

**Matrix:**

$W = [\omega_{ij}]$ the threat matrix implying the degree of threat between pairs of UAVs.

$H = [h_{ij}]$ the adjacency matrix of graph G.

$D = [d_{ij}]$ matrix that store the relative distance between UAVs.

**Variables:**

$l(G)$: The average distance of graph G.

$u_i$: Node that is chosen to be removed.

First, the UAV group is modelled, the UAV is replaced by a node, and the threatened UAV pairs are connected. Subsequently, the complex network can be represented by $G = (U, C)$ where $G$ is an undirected connected graph. We assume $n$ nodes and $m$ edges exist, and we represent $n$ UAVs and $m$ threats among them, respectively.

The threat coefficient $\omega_{ij} = v_{ij}/d_{ij}$ shows the degree of threat between $UAV_i$ and $UAV_j$, where $v_{ij}$ represents the projection value of the relative speed between $UAV_i$ and $UAV_j$ on its relative position, and $d_{ij}$ represents the relative position between $UAV_i$ and $UAV_j$.

**Definition 1.** *$\omega_{ij}$ is defined as follows:*

$$\omega_{ij} = \begin{cases} \dfrac{\vec{v}_{ij} \cdot \vec{d}_{ij}}{|\vec{d}_{ij}| \cdot |\vec{d}_{ij}|}, & \dfrac{\vec{v}_{ij} \cdot \vec{d}_{ij}}{|\vec{d}_{ij}| \cdot |\vec{d}_{ij}|} > 0 \\ 0, & \dfrac{\vec{v}_{ij} \cdot \vec{d}_{ij}}{|\vec{d}_{ij}| \cdot |\vec{d}_{ij}|} > 0 \end{cases} \tag{11}$$

*The adjacency matrix of graph G is represented by $H = [h_{ij}]$; it contains $n$ rows and $n$ columns. The element $h_{ij}$ is defined as follows:*

$$h_{ij} = \begin{cases} 1, & 0 < d_{ij} < \infty \\ 0, & d_{ij} = \infty \end{cases} \tag{12}$$

*The network cohesion is defined below. The degree of network cohesion depends on the connectivity of each node in the network, calculated using the arithmetic mean of the shortest distance between all pairs of nodes. The degree of network cohesion is still related to the density of the network; this is because in the UAV network, the more the number of UAVs, the greater is the threat of the entire network and the greater is the network cohesion.*

*The cohesion of graph G is defined as follows:*

$$\partial[G] = \frac{\omega}{n \cdot l} = \frac{\sum\limits_{i \neq j \in U} \omega_{ij}}{n \cdot \frac{\sum\limits_{i \neq j \in U} d_{ij}}{n(n-1)}} = \frac{(n-1) \cdot \sum\limits_{i \neq j \in U} \omega_{ij}}{\sum\limits_{i \neq j \in U} d_{ij}} \tag{13}$$

The following equation shows the importance of node $u_i$ [38]:

$$IM(u_i) = 1 - \frac{\partial[G]}{\partial[G - u_i]} = 1 - \frac{\frac{\omega_n}{n \cdot l(G)}}{\frac{\omega_{n-k_i}}{(n-k_i) \cdot l(G-u_i)}} = \frac{n \cdot l(G) \cdot \omega_{n-k_i} - (n - k_i) \cdot l(G - u_i) \cdot \omega_n}{n \cdot l(G) \cdot \omega_{n-k_i}} \tag{14}$$

Because $l(G - u_i) < l(G)$, $k_i > 1$, $\omega_n < \omega_{n-k_i}$, the following inequality $0 < IM(u_i) < 1$ can be achieved.

The algorithm is as follows: [38]:

Input: $H, W$

Output: $IM$

Step 1: Calculate the shortest relation distance matrix $D = [d_{ij}]$ for all pairs of UAV nodes;

Step 2: Calculate the initial cohesion of the UAV network;

Step 3: Assess the node importance of all UAV nodes.

Step 4: Calculate the network cohesion $\partial[G - u_i]$ after the node $u_i$ shrinks by calculating the shortest distance matrix $D_i$ and the threat matrix $W_i$ between the node pairs after contraction, thereby obtaining $IM(u_i)$.

As shown in the algorithm steps above, the time complexity of the whole algorithm depends on the calculation of the shortest distance matrix $D$ between all pairs of nodes after the node shrinks, because the calculation of the threat matrix $W_i$ only requires the deletion of the nodes around the current node $u_i$. Because the time complexity of the Floyd algorithm is $O(n^3)$, the complexity of the algorithm will reach $O(n^4)$, owing to the need to shrink the operation of each node. The algorithm for calculating $D_i$ given below is based on the initial matrix; the calculation can be completed with a small number of operations, and the time complexity of the entire algorithm will reduce to $O(n^3)$.

The direct distance matrix $S = [s_{ij}]_{n \times n}$ can be obtained by the adjacency matrix $H = [h_{ij}]$.

At this point, the change law of the shortest distance $d'_{pq}$ relative to $d_{pq}$ between any pair of nodes $(u_p, u_q) \in U \times U$ in the UAV network is as follows:

(1)　If $u_p \neq u_i$ and $u_q \neq u_i$:

　　① 　If $d_{pi} + d_{iq} = d_{pq}$, it is implied that node $u_i$ is on the shortest path between node $u_p$ and node $u_q$; then, $d'_{pq} = d_{pq} - 1$;

　　② 　If $d_{pi} + d_{iq} \geq d_{pq} + 1$, it is implied that node $u_i$ is not on the shortest path between node $u_p$ and node $u_q$; then $d'_{pq} = d_{pq}$.

(2)　If only one node $u_i$ exists between node $u_p$ and node $u_q$, then $d'_{pq} = d_{pq} - 1$;

(3)　If $u_p = u_q = u_i$, then $d'_{pq} = d_{pq} = 0$.

### 3.3. Direction of Collision Avoidance

Most of the CA systems use the upward or downward strategy to avoid collision. The sense selection can reduce the domino effect. For instance, two new encounters will occur if the key UAV descends, and three will occur if it climbs. The implementation of airborne CA systems II (ACAS II) [39] indicates that it is effective and safe to perform resolution in the vertical direction; therefore, the method of this paper only considered the vertical change in direction. As shown in Figure 3, the key node direction choosing method was as follows.

To generate a suitable direction for CA, we expect the UAV cluster system to minimize the cost and to choose the fastest method to reach the target, and we expect the domino effect to be as small as possible in the CA process. The objective function for direction choosing should include three elements:

(1)　The Number of Intruder UAV Designated Directions

$$Z_k(m) = \sum_{i=i}^{n} \text{sgn}(x_k(i)), \quad x_k(i) = \begin{cases} 1, & z_i > z_k \\ -1, & z_i \leq z_k \end{cases} \tag{15}$$

(2)   Network Robustness

Complex networks rely on their robustness, i.e., the ability of a network to maintain its connectivity when a fraction of its vertices is damaged, for their function and performance [40]. In the context of this new conflict model, it is important to improve reliability as much as possible. The robustness of the edge connection is defined as follows:

$$R_k(m) = R\left(\frac{I_m}{N_m}\right) = \frac{M_m}{N_m - I}, \; M = \frac{\sum\limits_{i=1}^{n}\sum\limits_{j=1}^{n} \omega_{ij}}{2}, \; \omega_{ij} = \frac{v_{ij}}{d_{ij}} \tag{16}$$

(3)   Connected Component of the Network

$$C_k(m) = C(\frac{I_m}{N_m}) = \frac{C_m}{N_m - I_m} \tag{17}$$

$N_m$ stand for N nodes in the network at time $m$, $I_m$ is the number of nodes that are rejected in the network at time $m$, and $C_m$ represents the number of sub-maps of the network.

In summary, the objective function at time $m$ consists of three elements and is represented as follows:

$$O_k(m) = \omega_1 Z_k(m) + \omega_2 R_k(m) + \omega_3 C_k(m) \tag{18}$$

In this formula, $\omega_1$, $\omega_2$, and $\omega_3$ are the corresponding weight coefficients of the three indicators. Different weights satisfy the requirements of different scenarios of CA. For instance, if $\omega_2 < \omega_3$, it is implied that the connected component is more important than the robustness of the network. It is noteworthy that a normalization process is required in this computation.

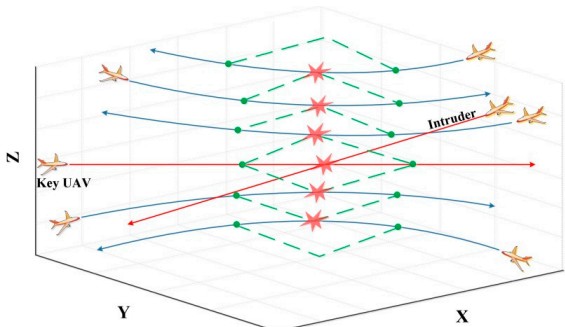

**Figure 3.** Conceptual graph for sense selection.

Assuming that the direction is amended to avoid the collision in the period $[m, m + N - 1]$ and the corresponding position and pitch angle sequences of $UAV_k$ are, respectively,

$$p_k = [p_k(m), p_k(m+1), \dots, p_k(m+N-1)]^T$$

$$\psi_k = [\psi_k(m), \psi_k(m+1), \dots, \psi_k(m+N-1)]^T$$

the following constraints of the optimization process should be satisfied:

$$q(\varphi_k(m + \Delta t)) = \begin{bmatrix} 1 \\ -1 \end{bmatrix}(\varphi_k(m + \Delta t) - \varphi_k(m + \Delta t)) - \begin{bmatrix} B \\ -A \end{bmatrix} \le 0$$

$$e_k(p_k(m + \Delta t)) = \begin{bmatrix} x_k(m + \Delta t) - [x_k(m) + v_k \cos(\varphi_k(m)) \cos \theta_k \cdot \Delta t] \\ y_k(m + \Delta t) - [y_k(m) + v_k \cos(\varphi_k(m)) \sin \theta_k \cdot \Delta t] \\ z_k(m + \Delta t) - [z_k(m) + v_k \sin \varphi_k \cdot \Delta t] \\ \varphi_k(m + \Delta t) - [\varphi_k(m) + \omega_k(m) \cdot \Delta t] \end{bmatrix} = 0 \tag{19}$$

In the formula above, $q(\varphi_k(m + \Delta t))$ implies that the direction change should be in the range of (A,B). $e_k(p_k(m + \Delta t))$ indicates that the pitch angle at time $m + \Delta t$ is determined by the global information at time $m$ and the constrains of the CA system. By repeating the optimization procedure, the complete amending trajectory is generated.

The goal of this new model is to decompose the network as quickly as possible. This implies that after the critical node is cleared, the robustness of the connection and the number of connected components should be as low as possible. This implies that the UAVs are leaving each other and the risk of crashes is becoming smaller. Meanwhile, this model also considers the number of intruder UAVs of the corresponding direction, implying that a lesser domino effect will occur because the density of the intruder UAV is small due to the sense selected by the model. In the simulation step, the new model obtains the key nodes and evaluates the robustness of the connections or the number of connected components when different choices are made by the critical nodes. The key nodes will select the course of change and form a new network in the next simulation step. The new model will again select the key node. The model will monitor multiple UAV systems to determine if a TA alert is present; if the model stops searching for critical nodes, multiple UAV systems will be considered safe.

It is noteworthy that this method is a solution to the deadlock problem. The reasons for solving the deadlock problem are as follows: During the entire simulation period, the threat detection between real-time UAVs is performed and the network is built in real time; the key nodes are selected, and the direction of CA is selected. Therefore, the deadlock problem can be avoided.

The algorithm proposed in this paper has a simple logical structure and strong stability. Decision calculations consume very little time and meet real-time requirements. The algorithm can be applied to engineering practice due to the following reasons. Firstly, the method we proposed is designed from the perspective of scheduling. Our innovation was to abstract the collision avoidance problem of the UAV to the network's collapse problem, aiming to solve the collision avoidance problem by using network theory. Second, the proposed method allows the UAV to perform only one maneuver—that of the flight level change. This maneuver is used in the traffic collision avoidance system (TCAS), and it is simple but very effective. As such, it can definitely be applied to the UAV collision avoidance system. The algorithm is shown below as Algorithm 1.

---

**Algorithm 1** Novel Model Based on Key Node Research

---

**Input:** Initial position, target position, initial velocity of the UAVs, the number of UAVs N, the simulation step length S
**Output:** Trajectories of all the UAVs R
**for** j = 1:S **do**
**for** i = 1:N **do**
　　Calculate whether a TA is issued;
　　**while** TA is issued **do**
　　Construct the network of the UAVs involved;
　　Calculate the relative distance and relative velocities of all the involved UAVs;
　　Select the key node;
　　Calculate the weight (relevant to the number of intruders, robustness and connected component);
　　Choose the optimal directions for the key node;
　　Update the current direction of the key node;
　　**if** no collision **then**
　　break;
　　**else**
　　R[i][j] = the coordinate of the step j of UAV i;
　　Calculate the state of all the UAVs involved in the scenario;
　**if** all the UAVs have reach the safe area **then**
　　break;
　return R;

---

## 4. Simulation and Results

We conducted two different simulation experiments on nine UAV scenarios for two different key node selection methods. The cruise speed set by the UAVs was 10 m/s. The minimum separation distance was set to 35 m. The simulation step size was set to 1 s. The pitch angle ranged from −0.785 to 0.785. It was assumed that each UAV can sense the global information and, as such, simulation calculation time can be ignored. The emulation code was run on a T450 laptop(Lenovo Thinkpad T450 Laptop, Lenovo, Beijing, China) with an Intel i7 processor of 2.6 GHz and 8 GB of RAM.

### 4.1. A Case Scenario

Figure 4a shows the UAV trajectory under the first key node selecting method, which is barely based on degree. In this scenario, the first time a TA appears is when $UAV_1$ and $UAV_9$ collide with each other. After the analysis of the CA system, we discovered that when avoiding the imminent collision, some potential collisions may occur. Therefore, when the TA issues a warning at 18:15:22 for the first time, the system selects $UAV_1$ as the key node and instructs $UAV_1$ to descend to avoid a conflict. As the simulation step progresses, $UAV_1$ and $UAV_9$ burst into each other's collision volumes. This time, $UAV_1$ becomes the key node, and a climb instruction is sent to $UAV_1$ to avoid collision. The result in Figure 4b demonstrates the conflict resolution maneuver and that the difference in this method arises because the key node in the same scenario is different. When selecting the key node by the node contracting method, the system selects the $UAV_9$ as the key node and instructs $UAV_9$ to descend to avoid conflict. These UAVs form a decentralized CA situation. This can be explained by the fact that the robustness of the network in this area has been minimized. The intuitive image is that three UAVs have spread out and no cross routes have appeared, thus avoiding the domino effect.

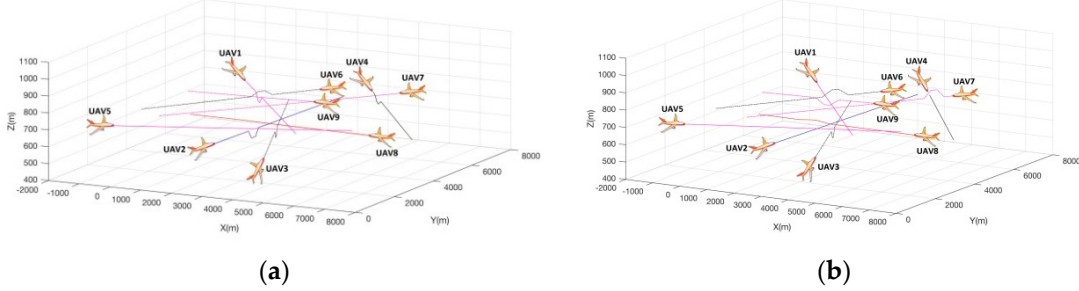

**Figure 4.** Trajectory (**a**) with key node selecting based on degree and (**b**) with key node selecting based on the node contraction method.

Table 2 summarizes the trajectory of $UAV_1$ and $UAV_9$ (the waypoints are recorded every 4 s for simplicity) in the same scenario under two different key node selecting methods. The UAV starts to change its angle at 18:15:23 and returns to the original trajectory at 18:17:23.

From this simulation, the results indicate that the CA system can facilitate the UAV cluster to effectively avoid collision in both scenarios. The figure illustrates the relative distance of each pair of UAVs that are involved in a collision scenario involving nine UAVs and four conflicts among them. It is noteworthy that the distance between $UAV_i$ and $UAV_j$ is defined by $d_{ij}$. Evidently, the separation of each pair of UAV is never intruded upon, according to Figure 5. For instance, in Scenario 1, $UAV_1$ encounters $UAV_6$ and $UAV_9$, and the distances between them are always larger than the separation distance even at the closest point such that the safety flight is guaranteed, and it is the same in Scenario 2.

**Table 2.** Waypoints of partial trajectory.

| Time | ID | Scenario 1 | | | ID | Scenario 2 | | |
| | | Amended Trajectory | | | | Amended Trajectory | | |
| | | X(m) | Y(m) | Z(m) | | X(m) | Y(m) | Z(m) |
|---|---|---|---|---|---|---|---|---|
| 18:15:19 | UAV1 | 876.27 | 6123.72 | 695.67 | UAV9 | 1762.27 | 5756.46 | 688.52 |
| 18:15:23 | UAV1 | 904.54 | 6095.45 | 695.53 | UAV9 | 1722.34 | 5754.09 | 688.48 |
| 18:15:27 | UAV1 | 919.23 | 6080.77 | 692.11 | UAV9 | 1695.73 | 5752.51 | 685.49 |
| 18:15:31 | UAV1 | 933.91 | 6066.08 | 688.69 | UAV9 | 1669.11 | 5750.93 | 682.51 |
| 18:15:35 | UAV1 | 948.60 | 6051.40 | 685.27 | UAV9 | 1642.50 | 5749.35 | 679.53 |
| 18:15:39 | UAV1 | 963.28 | 6036.71 | 681.85 | UAV9 | 1615.89 | 5747.76 | 676.55 |
| 18:15:43 | UAV1 | 977.97 | 6022.03 | 678.43 | UAV9 | 1589.28 | 5746.18 | 673.57 |
| 18:15:47 | UAV1 | 992.65 | 6007.34 | 675.02 | UAV9 | 1562.66 | 5744.60 | 670.58 |
| | UAV9 | 1482.76 | 5739.86 | 688.48 | UAV1 | 1074.25 | 5925.74 | 695.53 |
| 18:15:51 | UAV1 | 1007.33 | 5992.66 | 671.60 | UAV9 | 1536.05 | 5743.02 | 667.60 |
| | UAV9 | 1442.83 | 5737.48 | 688.48 | UAV1 | 1102.53 | 5897.46 | 695.53 |
| 18:15:55 | UAV1 | 1022.02 | 5977.97 | 668.18 | UAV9 | 1509.44 | 5741.44 | 664.62 |
| | UAV9 | 1402.90 | 5735.11 | 688.48 | UAV1 | 1130.81 | 5869.18 | 695.53 |
| 18:15:59 | UAV1 | 1036.70 | 5963.29 | 664.76 | UAV9 | 1482.83 | 5739.86 | 661.64 |
| | UAV9 | 1362.97 | 5732.74 | 688.48 | UAV1 | 1159.10 | 5840.89 | 695.53 |
| 18:16:03 | UAV1 | 1051.39 | 5948.60 | 661.34 | UAV9 | 1456.21 | 5738.28 | 658.66 |
| | UAV9 | 1323.04 | 5730.37 | 688.48 | UAV1 | 1187.38 | 5812.61 | 695.53 |
| 18:16:07 | UAV1 | 1066.07 | 5933.92 | 657.92 | UAV9 | 1429.60 | 5736.70 | 655.67 |
| | UAV9 | 1283.11 | 5727.99 | 688.48 | UAV1 | 1215.67 | 5784.32 | 695.53 |
| 18:16:11 | UAV1 | 1080.76 | 5919.23 | 654.50 | UAV9 | 1402.99 | 5735.12 | 652.69 |
| | UAV9 | 1243.18 | 5725.62 | 688.46 | UAV1 | 1243.95 | 5756.04 | 695.53 |
| 18:16:15 | UAV1 | 1105.63 | 5894.36 | 653.54 | UAV9 | 1366.39 | 5732.94 | 651.91 |
| | UAV9 | 1203.25 | 5723.25 | 688.48 | UAV1 | 1272.24 | 5727.75 | 695.53 |
| 18:16:19 | UAV1 | 1133.90 | 5866.09 | 653.43 | UAV9 | 1326.46 | 5730.57 | 651.86 |
| 18:16:23 | UAV1 | 1162.16 | 5837.83 | 653.27 | UAV9 | 1286.54 | 5728.20 | 651.82 |
| 18:16:27 | UAV1 | 1190.43 | 5809.56 | 653.13 | UAV9 | 1246.61 | 5725.83 | 651.77 |
| 18:16:31 | UAV1 | 1218.70 | 5781.29 | 652.99 | UAV9 | 1206.68 | 5723.45 | 651.72 |
| 18:16:35 | UAV1 | 1233.38 | 5766.61 | 656.41 | UAV9 | 1180.07 | 5721.87 | 654.70 |
| 18:16:39 | UAV1 | 1248.07 | 5751.92 | 659.82 | UAV9 | 1153.46 | 5720.29 | 657.68 |
| 18:16:43 | UAV1 | 1262.75 | 5737.24 | 663.24 | UAV9 | 1126.84 | 5718.71 | 660.67 |
| 18:16:47 | UAV1 | 1277.44 | 5722.55 | 666.66 | UAV9 | 1100.23 | 5717.13 | 663.65 |
| 18:16:51 | UAV1 | 1292.12 | 5707.87 | 670.08 | UAV9 | 1073.62 | 5715.55 | 666.63 |
| 18:16:55 | UAV1 | 1306.81 | 5693.18 | 673.50 | UAV9 | 1047.01 | 5713.97 | 669.61 |
| 18:16:59 | UAV1 | 1321.49 | 5678.50 | 676.92 | UAV9 | 1020.39 | 5712.39 | 672.59 |
| 18:17:03 | UAV1 | 1336.18 | 5663.81 | 680.34 | UAV9 | 993.78 | 5710.80 | 675.58 |
| 18:17:07 | UAV1 | 1350.86 | 5649.13 | 683.75 | UAV9 | 967.17 | 5709.22 | 678.56 |
| 18:17:11 | UAV1 | 1365.55 | 5634.44 | 687.17 | UAV9 | 940.56 | 5707.64 | 681.54 |
| 18:17:15 | UAV1 | 1380.23 | 5619.76 | 690.59 | UAV9 | 913.94 | 5706.06 | 684.52 |
| 18:17:19 | UAV1 | 1394.92 | 5605.07 | 694.01 | UAV9 | 887.33 | 5704.48 | 687.50 |
| 18:17:23 | UAV1 | 1416.39 | 5583.60 | 695.65 | UAV9 | 850.73 | 5702.31 | 688.21 |

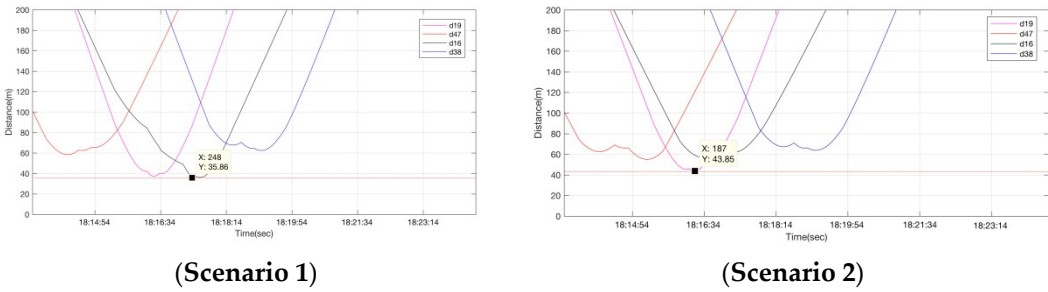

(**Scenario 1**)											(**Scenario 2**)

**Figure 5.** Minimal distance of UAVs for two scenarios.

Because the algorithm implements a single-machine change in direction, when the UAV does not reach the pre-set CA altitude, the relative distance of the UAV will be reduced slightly; however, it will not collide because it is at the predicted moment of collision. The UAV has already flown to a safe altitude; therefore, although a close trend may occur during CA, the UAV can maintain a safe relative distance in the subsequent flight path.

Figure 6 depicts the situation during the final CA of the UAV group. The connection represents the potential threat between the UAVs. The connection method is the same as the logic of building the network. It is related to the relative distance between nodes and the relative speed between the nodes. The network node can identify key nodes and subsequently select the most appropriate CA direction according to the network properties.

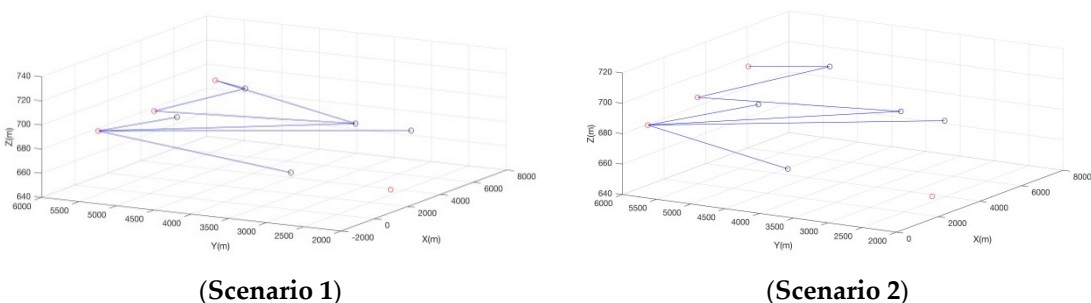

(**Scenario 1**)											(**Scenario 2**)

**Figure 6.** Distribution of UAVs during a threat in two scenarios.

We defined a new amount to measure the effect of UAVs when selecting key nodes using different methods. We defined this amount as the residual threat factor. The meaning of this quantity is the measure of the sum of threats to the UAVs in a certain area centered on the critical node after a conflict avoidance maneuver. That is, it evaluates the safety of the UAV in a certain area centered on the point after performing CA maneuver. Therefore, the threat coefficients of the UAVs in the area are added, and different key node selections are compared. The residual threat coefficient under the algorithm compares the advantages and disadvantages of the key node selection method. The residual threat factor (RTF) can be defined as follows:

$$\text{RTF}_t = \frac{\sum\limits_{i,j=1}^{N} t_{ij}^k}{N} \tag{20}$$

To depict the overall threat during the CA procedure, an average residual threat factor was defined as follows:

$$\overline{\text{RTF}} = \sum_{t \in trt} \text{RTF}_t, \ \text{TA}_{t \notin trt} = 0 \tag{21}$$

Table 3 shows that the $\overline{\text{RTF}}$ of the degree method was 0.0126, while the $\overline{\text{RTF}}$ of the node contraction method was 0.0093. This implies that the node contraction method has an average threat level lower than the degree method.

**Table 3.** $\overline{\text{RTF}}$ (residual threat factor) of two key node selecting approaches in two scenarios.

| Key Node Selecting Approach | $\overline{\text{RTF}}$ |
| --- | --- |
| Node selection based on degree | 0.0126 |
| Node contraction method | 0.0093 |

### 4.2. Further Investigation

This section discusses the primary results obtained using the CN algorithm based on two different key node selecting algorithms, as well as those obtained by the randomly chosen key node. It also reports the statistical comparison between those methods.

To validate the performance of the CN, a set of 500 different scenarios were grouped by the UAV number, from 24 to 48 within 100 tests. The extra consuming time is a frequently used performance metric to measure the different performance metrics that are used to measure the different performances of CA system algorithms.

Figure 7 shows the extra consuming time of the CN algorithm compared with the randomly selected 'sense.'

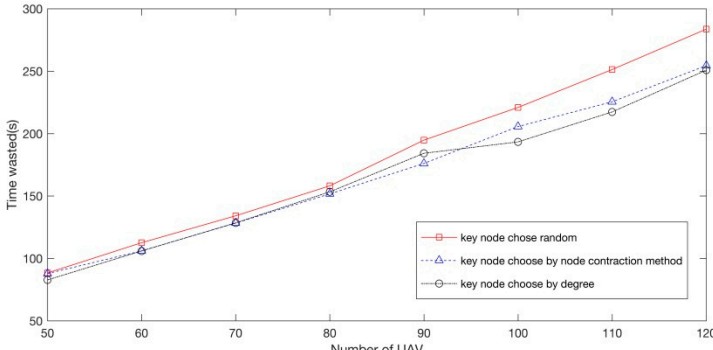

**Figure 7.** Consuming time of the complex network (CN) algorithm and random approach.

The simulation result presented herein clearly demonstrates the good performance of the CN algorithm for solving high-density CA problems, owing to the potential threat being solved preferentially and reducing the domino conflict. This shows that the wasted time increases if the number of UAVs increases, because more encounters will occur if more UAVs are cruising in the same limited airspace. This also shows the extra consuming time increases faster than the UAV density, and the difference between the extra consuming time under the CN algorithm and the extra consuming time under the random algorithm increases as the density of the UAV increases. However, while the key node is chosen randomly, the wasted time is always larger than that of the CN algorithm where the key node is chosen based on the degree or node contraction methods. Under the middle density circumstance (90 UAVs), the key node selection method by degree consumes more time than the key node selecting method by node contraction, because the node contraction method can obtain more potential threat information and can more accurately. choose the most threatened node The opposite applies for the high airspace density (more than 100 UAVs), because the airspace is sufficiently clouded such that the most intuitive method can more efficiently handle this type of situation.

We made a comparison of our proposed CN algorithm with the satisficing game theory-based algorithm (SGTA) [41], the reactive inverse proportional navigation(PN) algorithm (RIPNA) [42] and the geometric optimization model algorithm (GOM) [43], which is capable of dealing with large scale UAV cluster collision avoidance scenarios. This paper compared the computational time of different methods when the scene setting for each method was the same. The results for scenarios consisting of 20 UAVs, 40 UAVs and 60 UAVs are shown in Table 4.

**Table 4.** Time consuming comparison of the novel algorithm.

| Number of Aircrafts | Time Taken (s) | | | | |
|---|---|---|---|---|---|
| | SGTA | RIPNA | GOM | Complex Network Based on Degree | Complex Network Based on Node Contraction |
| 20 | 673 | 68 | 27 | 3 | 3 |
| 40 | 1711 | 200 | 68 | 9 | 10 |
| 60 | 3009 | 396 | 189 | 20 | 26 |

Evidently, the computational efficiency of the CN algorithm is higher than the other three algorithms, and it is worth mentioning that the average consuming time did not exponentially increase. Table 4 shows that this method can solve the collision avoidance problem in the case of high-density traffic airspace within an acceptable range.

## 5. Conclusions and Future Work

We herein proposed a CA method based on CNs for a group of UAVs in a local airspace. This method used the CN theory to synchronize the trajectory of the UAV in the global scope as quickly as possible to achieve CA. The collision avoidance method consisted of two different algorithms: The key node selection algorithm and the collision direction-selection algorithm. These two sub-algorithms formed a local space range that guaranteed the threat moment when the UAV groups met. The inner safety of the UAV group was the optimal core algorithm of the CA system. The key node selection algorithm constructed the key node selection strategy through the state representation and conflict detection logic of the UAV. The direction selection algorithm selected the threat cancellation scheme based on the minimum robustness principle working with the node selection algorithm. The two algorithms jointly applied a state-of-the-art CN theory to provide a method for resolving threats that occur when a group of unmanned aircrafts meets, by analyzing various states of the UAV group in the local airspace. In this paper, we compared the collision avoidance effects of UAVs under key node selection methods based on random selection and node contraction methods. The experimental results show that the key node selection strategy based on the node contraction method has better collision avoidance effects. The computational efficiency of the CN algorithm is higher than the satisficing game theory-based algorithm (SGTA), reactive inverse PN algorithm (RIPNA) and geometric optimization model algorithm (GOM), and it is worth mentioning that the average consuming time did not exponentially increase.

Future research will focus on the following aspects: (1) Using appropriate heuristic algorithms to improve the efficiency and speed of algorithm implementation; (2) conducting a more comprehensive study of the safety indicators of UAVs and integrating them into the existing ones, thus establishing an anti-collision system to enhance the safety of the system for the UAV group; (3) considering more complicated disturbance situations (such as strong wind) to accommodate more intensive and complex situations; (4) integrating UAVs into general purpose aviation operations in a non-segregated airspace.

**Author Contributions:** Conceptualization, Y.H. and J.T.; methodology, Y.H.; software, Y.H.; validation, Y.H.; formal analysis, Y.H.; investigation, Y.H.; resources, Y.H., J.T. and S.L.; data curation, Y.H.; writing—original draft preparation, Y.H.; writing—review and editing, Y.H. and J.T.; visualization, Y.H.; supervision, J.T. and S.L.; project administration, J.T. and S.L.; funding acquisition, J.T. and S.L.

**Funding:** This research received no external funding.

**Conflicts of Interest:** The authors declare no conflict of interest.

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
