# Peer review of "Cooperative Multi-UAV Collision Avoidance Based on a Complex Network"

_applsci, doi:10.3390/app9193943_

Round 1

Reviewer 1 Report

N/A.

Reviewer 2 Report

Summary: "This article focuses on the cooperative Multi-UAV collision avoidance system. The proposed approach allows the UAV to perform only one maneuver that includes flight level change. The novel UAV conflict resolution is divided into two steps, corresponding to the key node selection (KS) algorithm and sense selection (SS) algorithm."

My comments are given below:

--What are the failure rate of the proposed approach and how much accuracy is observed in the proposed model?

--The authors should include recent works in Table 1 as well as include their proposed work to show exact differences.

-- One major concern is that an optimal solution will work better than the random selection. Thus, why authors compare their work with random selection, which otherwise should not be used with UAVs.

--Proof of optimality and complexity analysis can be discussed for the given algorithm.

--Why does the approach behave linearly after a certain number of UAVs. The decision time should not be much affected by the total UAVs, rather it should be affected by the number of surrounding UAVs.

-- The intruder model should be explained with more details including the role and capabilities of an intruder and how many and how much control can be prevented by using the proposed approach.

-- The literature needs further improvements especially considering the fact that there are recent solutions proposed on the collision avoidance in a multi-UAV setup. Based on this, I would recommend authors to consider the following articles as well as search for additional related solutions:
https://www.mdpi.com/1996-1073/12/8/1551
https://ieeexplore.ieee.org/abstract/document/8337908/
https://link.springer.com/article/10.1007/s10846-018-0929-y

Round 2

Reviewer 2 Report

No further comments. The authors provided point to point answers to my queries.

This manuscript is a resubmission of an earlier submission. The following is a list of the peer review reports and author responses from that submission.

Round 1

Reviewer 1 Report

Comments to the authors:

The paper entitled “Cooperative Multi-UAV Collision Avoidance Based on Complex Network” studies the problem of conflict detection and resolution in cooperative unmanned aerial vehicle (UAV) clusters.

However, I have some concerns as follows:

Technical aspect:

-       There is a lack of explanation of the rationale used in the study.

-       Each part is independent of each other and no relationship can be found.

-       According to the respected authors, the paper is about the “Cooperative Multi-UAV Collision Avoidance Based on Complex Network”. However, it seems that they studied the problem of conflict detection and resolution. Please provide the rationale.

-  The literature review section at the beginning is superficial and does not represent the state-of-the-art both considering “conflict detection and resolution” and “collision avoidance”. For example, articles related to the collision avoidance of multi-agent, etc.

Olfati-Saber, R., 2006. Flocking for multi-agent dynamic systems: algorithms and theory. IEEE Transactions on Automatic Control51(3), pp.401-420.

Rezaee, H. and Abdollahi, F., 2011, July. Mobile robots cooperative control and obstacle avoidance using potential field. In 2011 IEEE/ASME International Conference on Advanced Intelligent Mechatronics (AIM) (pp. 61-66). IEEE.

Rezaee, H. and Abdollahi, F., 2013. A decentralized cooperative control scheme with obstacle avoidance for a team of mobile robots. IEEE Transactions on Industrial Electronics61(1), pp.347-354.

Nguyen, T., La, H.M., Le, T.D. and Jafari, M., 2016. Formation control and obstacle avoidance of multiple rectangular agents with limited communication ranges. IEEE Transactions on Control of Network Systems4(4), pp.680-691.

Duguleana, M. and Mogan, G., 2016. Neural networks based reinforcement learning for mobile robots obstacle avoidance. Expert Systems with Applications62, pp.104-115.

-       I would recommend the respected authors to study the following work regarding the path planning:

LaValle, S.M., 2006. Planning algorithms. Cambridge university press.

-       Abstract, Introduction, and Conclusion Sections need to be rewritten and reconstructed to improve the quality of the paper.

-       What are the main contributions of this paper in comparison to the following articles?

Huang, Y., Tang, J. and Lao, S.Y., 2019. Collision Avoidance Method for Self-organizing Unmanned Aerial Vehicle Flights. IEEE Access.

Lao, M. and Tang, J., 2017. Cooperative multi-UAV collision avoidance based on distributed dynamic optimization and causal analysis. Applied Sciences, 7(1), p.83.

Sun, J., Tang, J. and Lao, S., 2017. Collision avoidance for cooperative UAVs with optimized artificial potential field algorithm. IEEE Access, 5, pp.18382-18390.

Wan, Y., Tang, J. and Lao, S., 2019. Distributed Conflict-Detection and Resolution Algorithms for Multiple UAVs Based on Key-Node Selection and Strategy Coordination. IEEE Access, 7, pp.42846-42858.

-       Comparison with existing methodologies is required.

-        What is the computational complexity of the proposed method?

-       Can the proposed method be employed in real-time?

-       It would be nice if the authors could evaluate the performance of the system in noisy environment.

Presentation aspect:

-       The paper need improvement in presentation.

-       Linguistics, readability of the paper should be improved, and the authors should restructure the paper in order to have a smooth transition among the sections.

-       Avoid long sentences. Lengthy and convoluted sentences make the text hard to read.

-       Replace all the figures with the new ones with good quality.

-       Table 1 is hard to read. Please review the paper before final submission.